# Reducing Sequence Length by Predicting Edit Spans with Large Language Models

**Masahiro Kaneko**[1,2]    **Naoaki Okazaki**[2]
[1]MBZUAI
[2]Tokyo Institute of Technology
Masahiro.Kaneko@mbzuai.ac.ae    okazaki@c.titech.ac.jp

## Abstract

Large Language Models (LLMs) have demonstrated remarkable performance in various tasks and gained significant attention. LLMs are also used for local sequence transduction tasks, including grammatical error correction (GEC) and formality style transfer, where most tokens in a source text are kept unchanged. However, the models that generate all target tokens in such tasks have a tendency to simply copy the input text as is, without making needed changes, because the difference between input and output texts is minimal in the training data. This is also inefficient because the computational cost grows quadratically with the target sequence length with Transformer. This paper proposes predicting edit spans for the source text for local sequence transduction tasks. Representing an edit span with a position of the source text and corrected tokens, we can reduce the length of the target sequence and the computational cost for inference. We apply instruction tuning for LLMs on the supervision data of edit spans. Experiments show that the proposed method achieves comparable performance to the baseline in four tasks, paraphrasing, formality style transfer, GEC, and text simplification, despite reducing the length of the target text by as small as 21%. Furthermore, we report that the task-specific fine-tuning with the proposed method achieved state-of-the-art performance in the four tasks.

## 1 Introduction

Large Language Models (LLMs), including Chat-GPT[1] and Bard[2], have exhibited exceptional performance across a range of natural language processing (NLP) tasks and amassed a significant user base (Brown et al., 2020; Chowdhery et al., 2022; OpenAI, 2023). As performance gains are brought from the increases in model size (Kaplan et al., 2020; Wei et al., 2022; Zhao et al., 2023), LLMs

---

[1]https://chat.openai.com/
[2]https://bard.google.com/

are becoming larger and larger. However, the computational cost of inference is a severe bottleneck of many practical applications, especially when the number of parameters in an LLM is massive (Bender et al., 2021; Kraus et al., 2023).

Meanwhile, LLMs are also used for local sequence transduction tasks, such as paraphrasing, formality style transfer, Grammatical Error Correction (GEC), and simplification (Kaneko et al., 2022; Reif et al., 2022; Wu et al., 2023a; Wang et al., 2022; Kaneko and Okazaki, 2023), where only a small portion of the source text is edited. Most tokens in a source text are kept unchanged in these tasks. For example, the source text, *"Many years ago, the situation is different,"* and the target text, *"Many years ago, the situation was different,"* of the GEC task mostly share the common tokens except for the underlined tokens (*is* and *was*).

Existing methods of downstream tasks do not make use of the characteristics of local sequence transduction (Reif et al., 2022; Wu et al., 2023a; Wang et al., 2022), simply generating all target tokens. In this paper, we hypothesize that this treatment is disadvantageous in achieving high performance in terms of task accuracy and computational time. More specifically, it is inefficient to generate unchanged tokens (e.g. *Many, years, ago, the, situation, different*) in the previous example because the model must copy many source tokens only to increase the length of the target sequence.

This study proposes to predict a set of edit spans, which represent the changed parts of the target text relative to the source tokens. Omitting unedited tokens that occupy most of the target text, we can reduce the length of the target text and the inference time for local sequence transduction tasks. Figure 1 shows the process of creating a set of edit spans from source and target texts in GEC. First, we align tokens in the source and target texts to extract the edit locations and tokens and convert them into a set of edit spans. In the example shown

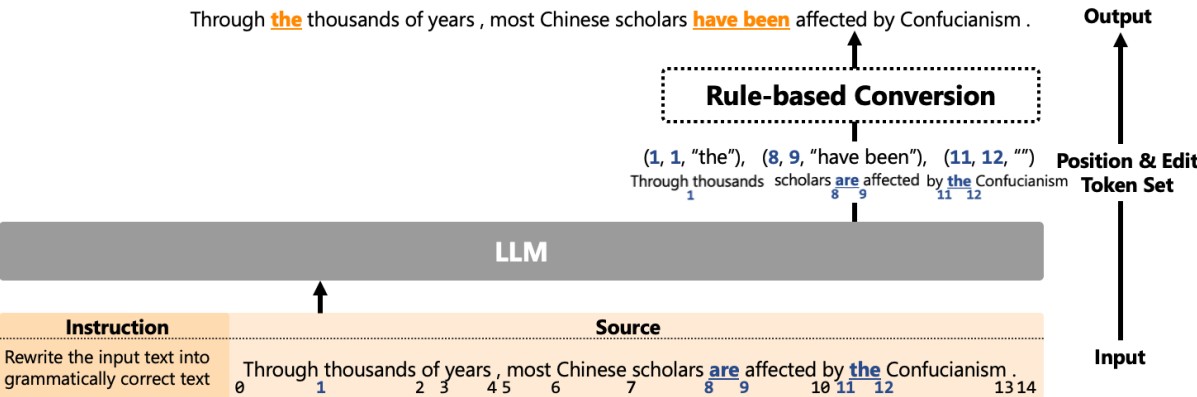

Figure 1: Inference of instruction tuned LLMs using edit spans. LLMs take instruction text and source text as input and output only the positions and tokens for rewriting. Rule-based conversion applies the outputted positions and tokens of the rewriting to the source text and produces the plaintext output.

in Figure 1, the edit spans (1, 1, "*the*"), (8, 9, "*have been*"), (12, 13, "") are created from the source text *"Through thousands of years, most Chinese scholars are greatly affected by the Confucianism."* and the target text *"Through the thousands of years, most Chinese scholars have been greatly affected by Confucianism."*. LLMs are fine-tuned using pairs of source text and edit spans with the instructions.

We conducted experiments on four local sequence transduction tasks: paraphrasing, formality style transfer, GEC, and simplification. The proposed method achieved comparable performance to the baseline that directly outputs the target text. In these tasks, the proposed method could reduce the sequence length on the target side by 32% on average and by as small as 21% in GEC. Furthermore, the proposed method with task-specific fine-tuning achieved state-of-the-art (SoTA) performance in the four tasks.

## 2 Edit Spans

### 2.1 Edit Span Extraction

To extract the editing locations and results of the source and target texts, we calculate the alignment between the tokens in each text. We use linguistical alignment, which incorporates linguistic information, to perform the alignment (Felice et al., 2016). Linguistical alignment is a method based on the Damerau-Levenshtein algorithm that aligns tokens by considering not only the distance between tokens but also the match of their lemma, part-of-speech, and character features, weighted accordingly. Taking into account the linguistic information of tokens, linguistical alignment is more

accurate compared to alignment methods that only use surface information. Furthermore, linguistic alignment merges token alignments using recursive rules to create alignments for multiple tokens, such as *"have been"* in Figure 1.

To indicate the edit position identified by the alignment, a 0 is assigned before the first token of the source text, and an index is sequentially assigned to the space after each token. When the length of the source text is $N$, $N$ is assigned after the last token. The edit span is represented by the tuple of the start position of the source text, the end position of the source text, and the token result after being edited.

There are three types of edit operations: insert, replace, and delete; we explain them using the example in Figure 1. The tuple (1, 1, "*the*") represents the operation to insert "*the*". In an insertion operation, both the start and end positions are set to the same position where a token is inserted in the source. The tuple stands for inserting "*the*" between the tokens located at the 1st position. The tuple (8, 9, "*have been*") presents the operation to replace "*are*" with "*have been*". By specifying the 8th and 9th positions of the source text, this tuple targets the "*are*" and rewrites them as "*have been*". The tuple (12, 13, "") represents the operation to delete "*the*". It points "*the*" by specifying the 12th and 13th positions in the source text. Because the target token after this edit operation is empty, this tuple corresponds to removing "*the*".

### 2.2 Instruction Tuning with Edit Spans

Instruction tuning fine-tunes LLMs by using natural language instructions describing a task (Wei

et al., 2021). Compared to the conventional fine-tuning that specializes the model for a specific task, instruction tuning aims for generalization to various tasks by training LLMs to respond well to many kinds of instructions. Therefore, instruction tuning is used for training many LLMs in an open-ended setting (Ouyang et al., 2022; Chung et al., 2022; Wang et al., 2022; Wu et al., 2023b). We use the description of local sequence transduction tasks as instructions to perform instruction tuning of LLMs. We provide the LLMs with instructions and source text, and train the LLMs to generate edit spans. When there are multiple edits, they are concatenated with commas like *"1 1 the, 8 9 have been, 12 13"*. When no editing is required in the source text, *"None"* is given as the gold text.

Recent LLMs are expected to have the ability to handle unknown tasks and various tasks, to achieve generality. It is important that learning through edit spans does not degrade the performance of tasks other than local sequence transduction tasks. Therefore, we add edit span data to the existing training data for instruction tuning, which includes various tasks, and fine-tune LLMs.

## 2.3 Conversion from Edit Spans to Output Text

To convert the edit spans output by LLMs into plaintext, we use a rule-based approach. If LLMs generate *"None"*, we use the source text as the final output text. Otherwise, we split the edit spans by commas and extract the edits. From each edit, we extract the starting position, ending position, and edited result. If LLMs generate edits in an incorrect format that do not include start or end positions or edits where the start or end positions exceed the source text range, we ignore them. To ensure that the token indices do not shift, we apply the edits to the source text in descending order of starting positions. This conversion is implemented by simple rules with a minimal computational cost.

## 3 Experiment Setting

### 3.1 Local Sequence Transduction Taskes

We conducted experiments on local sequence transduction tasks such as GEC, paraphrasing, formality style transfer, and simplification.

**GEC** We used NUCLE as the training data, CoNLL2013 (Ng et al., 2013) as the development data, and CoNLL2014 (Ng et al., 2014) as the evalu-

ation data. The dataset is comprised of essays composed by college students from the National University of Singapore, covering a broad spectrum of subjects, including environmental pollution, healthcare, and more. We used the $M^2$ score (Dahlmeier and Ng, 2012) as the evaluation metric. For GEC, we provide the instruction text *"Rewrite the input text into grammatically correct text."*.

**Paraphrasing** Quora published a dataset that includes more than 400K lines of potential question duplicate pairs[3]. Of these pairs, 150K question pairs were labeled as paraphrases. Only those labeled paraphrase question pairs are used as training, development, and test sets. We used BLEU-4 (Papineni et al., 2002), ROUGE-1, and ROUGE-2 (Lin, 2004) to evaluate LLMs, following previous research (Kumar et al., 2020; Meng et al., 2021; Li et al., 2022). For paraphrasing, we provide the instruction text *"Rewrite the input text into paraphrased text."*

**Style transfer** We used FST benchmark Grammarly Yahoo Answers Corpus (GYAFC) (Rao and Tetreault, 2018) for formality style transfer. GYAFC is a plain corpus that contains pairs of informal and formal sentences conveying the same meaning. It covers domains such as Entertainment & Music (E&M) and Family & Relationship (F&R). We utilized the corpus BLEU in NLTK (Bird and Loper, 2004) as described in Chawla and Yang (2020). For formality style transfer, we provide the instruction text *"Rewrite the input text into formal text."*

**Simplification** We used WikiSmall[4] (Zhu et al., 2010; Zhang and Lapata, 2017) as the training data and ASSET (Alva-Manchego et al., 2020) and TurkCorpus (Xu et al., 2016) as the evaluation data. We used SARI (Xu et al., 2016) to evaluate LLMs, which compares the generated text with the target text and calculates the average F1 score for addition, keep, and deletion operations. For text simplification, we provide the instruction text *"Rewrite the input text into simpler text."*

### 3.2 Open-ended Tasks

The rules of edit spans differ from the rules in the raw text, which could potentially have a negative impact on the performance of tasks other than local

---

[3]https://www.kaggle.com/c/quora-question-pairs

[4]https://github.com/XingxingZhang/dress

sequence transduction. By combining open-ended instruction tuning data and edit spans instruction tuning data, we can train LLMs and investigate their impact on other tasks as well.

We utilize the databricks-dolly-15k dataset[5] by randomly dividing it into 13K for training, 1K for development, and 1K for evaluation. databricks-dolly-15k is a publicly available dataset consisting of instructional records created by numerous Databricks employees. It covers various behavioral categories described in InstructGPT (Ouyang et al., 2022), such as brainstorming, classification, closed QA, generation, information extraction, open QA, and summarization. We sampled 3K instances for each of the tasks: GEC, paraphrasing, style transfer, and simplification, resulting in a total of 12K instruction instances. We fine-tuned LLMs using a combined dataset of all these instructions, totaling 25K instances.

We used BERTScore[6] (Zhang et al., 2019) as our evaluation metric. BERTScore is an evaluation method that measures the similarity between generated text and target text using contextual embeddings from pre-trained models. We utilized RoBERTa (Liu et al., 2019) (roberta-large[7]) as the BERTScore models.

## 3.3 Instruction Tuning Settings

We used the following four LLMs for our experiments: MPT (mpt-7b)[8] (Team, 2023), OPT (opt-6.7b)[9] (Zhang et al., 2022), LLaMA (llama-7b)[10] (Touvron et al., 2023), and BLOOM (bloom-7b1)[11] (Scao et al., 2022).

We used the code for instruction tuning from Stanford Alpaca (Taori et al., 2023) code[12] for instruction tuning. We set the number of epochs to 3 and used a batch size of 32. The learning rate was set to 2e-5, with a warmup rate of 0.03, and we employed a cosine learning rate schedule. These hyperparameters were determined following Stanford Alpaca. We report the average results of three models trained with different seeds for instruction tuning. We used four nodes, each containing eight

---

[5] https://huggingface.co/datasets/databricks/databricks-dolly-15k/viewer/databricks--databricks-dolly-15k
[6] https://github.com/Tiiiger/bert_score
[7] https://huggingface.co/roberta-large
[8] https://huggingface.co/mosaicml/mpt-7b
[9] https://huggingface.co/facebook/opt-6.7b
[10] https://github.com/facebookresearch/llama
[11] https://huggingface.co/bigscience/bloom-7b1
[12] https://github.com/tatsu-lab/stanford_alpaca

NVIDIA A100 GPUs. We used the code[13] for linguistical alignment provided by Felice et al. (2016).

**Baselines** We compare the results of the proposed method with the results of LLMs fine-tuned for instruction tuning using the target text as the ground truth instead of using edit spans. This comparison examines whether edit spans can reduce computational costs during inference without compromising performance.

## 4 Experiment

### 4.1 Performance on Local Sequence Transduction Tasks

To demonstrate the contribution of edit spans to performance improvement, we first compare the baseline performance with fine-tuned data using plain text. Table 1 shows the results of performance comparison between the baseline and the proposed method in the GEC, paraphrasing, style transfer, and simplification tasks. Out of 32 cases, performance improvement was observed in 19 cases, and edit spans contributed to the performance enhancement. Furthermore, it can be observed that the LLaMA trained with edit spans achieves the highest performance in most cases.

### 4.2 Reducing Text Length

We examine how much the fine-tuning of LLMs with edit span data reduced the length of the output text. Figure 2 shows the ratio of output text length to target text length when fine-tuned with plain data and edit span data, respectively, on the development data for each task. The proposed method successfully compresses the output text across all tasks, independent of the model used; it achieves text compression in the range of 21% in the most compressed cases and 41% even in the least compressed cases. In GEC, there are cases where grammatically correct text is provided as source text. In such cases, the model does not need to make any revisions and can simply output *"None"*, resulting in significant compression in GEC.

### 4.3 Performance on Open-ended Task

In open-ended tasks, the target texts are written in plain text, while edit spans introduce significant differences in text formatting. This misalignment in text representation may potentially impact the performance of open-ended tasks. Therefore, we aim

---

[13] https://github.com/chrisjbryant/errant

|  |  | GEC | Paraphrasing | Style transfer | Simplification |
|---|---|---|---|---|---|
| Plain | MPT | 68.0 | 37.9/66.5/47.1 | 78.9/81.2 | 46.3/41.1 |
|  | OPT | 65.7 | 35.2/63.2/45.4 | 75.0/77.2 | 43.7/40.5 |
|  | LLaMA | 68.2 | **39.3**/69.0/47.2 | **79.5**/81.0 | 48.0/41.9 |
|  | BLOOM | 66.4 | 37.0/66.4/46.1 | 78.2/79.9 | 45.0/41.0 |
| Edit spans | MPT | 68.5 | 38.2/66.7/47.1 | 78.2/81.3 | 46.6/41.3 |
|  | OPT | 66.2 | 34.1/61.2/43.9 | 75.6/77.9 | 43.9/40.3 |
|  | LLaMA | **69.1** | 39.0/**69.2**/**47.6** | 79.3/**81.2** | **48.3**/**42.0** |
|  | BLOOM | 65.8 | 37.2/66.1/46.3 | 78.0/80.3 | 44.8/40.7 |

Table 1: The performance of four LLMs fine-tuned with edit spans and plain data instructions on four local sequence transduction tasks. The **bold** values indicate the highest performance for each task. The underlined values indicate when edit spans exceed the baseline.

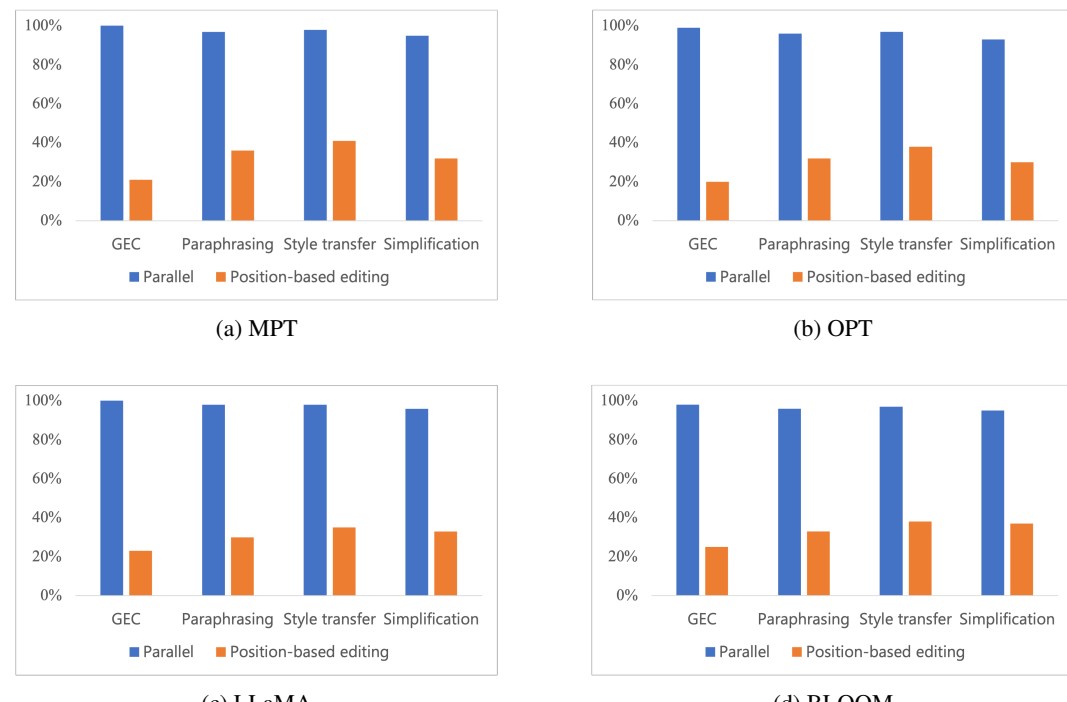

(a) MPT

(b) OPT

(c) LLaMA

(d) BLOOM

Figure 2: The ratio of output text length to target text length when MPT, OPT, LLaMA, and BLOOM are fine-tuned with plain data and edit span data, respectively.

to demonstrate that edit spans do not significantly degrade the performance of open-ended tasks.

Table 2 shows the scores for each LLM when using RoBERTa as BERTScore models on the 1K subset of the databricks-dolly-15k dataset, which was divided for evaluation. This indicates that the proposed method achieves efficient computational cost during inference without significantly sacrificing open-ended task performance.

To maintain performance in open-ended tasks, the proposed method combines data from both local sequence transduction tasks and open-ended tasks. To demonstrate the effectiveness of combining open-ended task data, we also investigate the open-ended task performance of instruction-tuned LLMs when solely trained on local sequence

transduction task data.

Table 3 demonstrates the performance difference on the 1K split of the databricks-dolly-15k dataset, evaluating LLMs trained on both open-ended task and local sequence transduction task data versus LLMs trained solely on local sequence transduction task data. The performance decreases when not using open-ended task data for training, both in terms of plain text and edit spans. This is likely because open-ended task data consists of plain text, while edit spans include totally different text formats, leading to a larger disparity in task requirements.

|          |        | BERTScore |
|----------|--------|-----------|
|          | MPT    | 81.5      |
| Plain    | OPT    | 79.3      |
|          | LLaMA  | 81.8      |
|          | BLOOM  | 79.9      |
|          | MPT    | 81.0      |
| Edit span| OPT    | 78.6      |
|          | LLaMA  | 81.3      |
|          | BLOOM  | 79.5      |

Table 2: Scores using BERTScore on the databricks-dolly-15k dataset, which was divided for evaluation.

|          |        | BERTScore diff. |
|----------|--------|-----------------|
|          | MPT    | -5.2            |
| Plain    | OPT    | -5.7            |
|          | LLaMA  | -4.4            |
|          | BLOOM  | -6.2            |
|          | MPT    | -8.1            |
| Edit span| OPT    | -8.6            |
|          | LLaMA  | -6.9            |
|          | BLOOM  | -7.6            |

Table 3: The performance difference between instruction tuned LLMs using local sequence transduction task and open-ended task datasets, and instruction tuned LLMs using only local sequence transduction task datasets.

## 4.4 The Accuracy of Edits Generated by the LLMs

Even if the edit span text is different, there are cases where the text is transformed by the rule, and the text matches. For example, in GEC, the model is given the input *"This technology could also be seen as invasion of human privacy."* and the model outputs *"7 9 invading"*. In this case, even with the alternate edit span text *"7 8 invading, 8 9"*, the conversion based on the rules would result in the same output text. However, this would increase the sentence length by the index, creating room for improvement in terms of computational cost. Therefore, we investigate how well edit span of the model matches the results using linguistic alignment.

First, we convert the edit spans generated by the model to plain text using rules. From the converted plain text and the source text, we create edit spans using linguistic alignment and calculate the percentage of agreement with the edit spans output by the model. Only when the start position $s$, end position $e$, and the edit token $r$ all match exactly is it considered a correct answer.

Table 4 shows the percentage of agreement between the edit spans output by the LLMs and the edit spans extracted by the linguistic alignment in the development data for each task. The proposed method achieves more than 90% agreement in 13 out of 16 settings. This indicates that LLMs are able to learn the extraction rules for linguistical alignment through instruction tuning.

## 4.5 Task-specific Fine-tuning

In the previous experiments, LLMs were trained by combining data from the four local sequence transduction tasks and the open-ended task. To explore the maximum potential performance of the proposed method, we fine-tune LLMs with task-specific focus using edit span data. We fine-tune LLMs for each task using all available training data. In this case, we specialize LLMs for specific tasks without the need for instruction texts. Therefore, we trained the LLMs by providing only the source texts as input.

We trained LLaMA, which showed the highest performance in the local sequence transduction tasks. We set the number of epochs to 2 and used a batch size of 32. The learning rate was set to 1e-5, with a warmup rate of 0.03, and we employed a cosine learning rate schedule. Following the exploration method described in Section 3.3, we determined the hyperparameters for our experiments.

Table 5 shows the results of performance comparison with existing studies on GEC, paraphrasing, style transfer, and simplification tasks. The proposed method outperforms existing studies by 1.8 points in GEC, 0.9, 1.2, and 2.3 points in paraphrasing, 1.9 and 1.3 points in style transfer, and 1.2 and 0.7 points in simplification tasks, respectively. Thus, the proposed method achieves the SoTA performance in all tasks. From these results, it can be concluded that edit spans are an effective method, even in task-specific fine-tuning scenarios.

## 4.6 Example of LLMs Output Using Edit Spans

Table 6 shows the output in CoNLL2013 for LLaMA using edit span and LLaMA outputting plain text. The normal model outputting plain text outputs 23 tokens, while the model using edit span outputs only 3 tokens. The output of the model using the edit span is a much shorter sequence than the original model that outputs plain text. Furthermore, LLaMA, which outputs in plain text, is unable to correct a grammatical error. In a local sequence transduction task, most tokens in the source

|       | GEC  | Paraphrasing | Style transfer | Simplification |
|-------|------|--------------|----------------|----------------|
| MPT   | 96.6 | 95.0         | 89.2           | 94.7           |
| OPT   | 93.3 | 91.9         | 88.8           | 92.7           |
| LLaMA | 99.0 | 96.2         | 92.6           | 95.4           |
| BLOOM | 94.2 | 92.5         | 89.4           | 93.5           |

Table 4: Agreement between edit spans generated by LLMs and edit spans extracted by linguistic alignment.

|                              | GEC      |
|------------------------------|----------|
| (Kaneko et al., 2020)        | 65.2     |
| (Omelianchuk et al., 2020)   | 66.5     |
| (Qorib et al., 2022)         | 69.5     |
| Edit span                    | **71.3** |

(a) $M^2$ scores on the CoNLL2014 dataset.

|                       | Paraphrasing        |
|-----------------------|---------------------|
| (Kumar et al., 2020)  | 38.0/68.1/45.7      |
| (Meng et al., 2021)   | 26.8/65.0/38.5      |
| (Li et al., 2022)     | 39.3/70.8/48.3      |
| Edit span             | **41.2/72.0/50.6**  |

(b) BLEU-4, ROUGE-1, and ROUGE-2 scores on the Quora dataset.

|                         | Style transfer    |
|-------------------------|-------------------|
| (Chawla and Yang, 2020) | 76.2/79.9         |
| (Lai et al., 2021)      | 76.5/79.3         |
| (Liu et al., 2022)      | 78.8/81.4         |
| Edit span               | **80.7/82.7**     |

(c) NLTK BLEU scores on the E&M and F&R datasets.

|                        | Simplification    |
|------------------------|-------------------|
| (Martin et al., 2020)  | 40.1/41.4         |
| (Martin et al., 2022)  | 44.2/42.6         |
| (Feng et al., 2023a)   | 47.9/41.8         |
| Edit span              | **49.1/43.5**     |

(d) SARI scores on ASSET and TurkCorpus datasets.

Table 5: Performance comparison with previous studies on GEC, paraphrasing, style transfer, and simplification tasks.

text and target text are common, and the model tends to learn just to copy the input tokens (Rastogi et al., 2016). Contrarily, our model that uses edit spans outputs only the edited parts. Thus simply copying the input is not an issue for our model.

## 5 Related Work

### 5.1 Efficient LLMs

Most of the methods for achieving efficient LLMs involve improving the memory complexity of self-attention mechanisms or enhancing the overall efficiency of the Transformer architecture (Tay et al., 2022; Loem et al., 2022). In the initial stages, the modifications made to self-attention focused on reducing the computational complexity by introducing sparsity in the attention matrix. This was accomplished by restricting the attention's scope to predetermined patterns, such as local windows and fixed stride block patterns (Liu et al., 2018; Qiu et al., 2020; Beltagy et al., 2020). A natural extension to the blockwise method is to connect these blocks via recurrence. Dai et al. (2019) introduced a mechanism of segment-level recurrence that establishes connections among multiple segments and blocks.

An expansion upon fixed, predetermined pat-terns is the utilization of learnable patterns. Models that incorporate learnable patterns aim to acquire the access pattern through data-driven methods. One crucial aspect of learning patterns is to establish a concept of token relevance and subsequently assign tokens to buckets or clusters (Vyas et al., 2020; Wang et al., 2021; Kitaev et al., 2020; Tay et al., 2020; Roy et al., 2021).

Another approach is to utilize a trainable side memory module capable of accessing multiple tokens simultaneously (Sukhbaatar et al., 2019; Ainslie et al., 2020; Beltagy et al., 2020). A prevalent example is the global neural memory, which can access the entire sequence. The global tokens function as a type of model memory, learning to gather information from the input sequence tokens.

Another method to enhance efficiency is by utilizing low-rank approximations of the self-attention matrix to improve computational performance (Wang et al., 2020), and to view the attention mechanism through kernelization (Choromanski et al., 2020; Peng et al., 2021). Sparse models selectively activate a fraction of the parameters, resulting in an improved parameter to FLOPs ratio in general (Fedus et al., 2022).

As a way to reduce the length of the text, Cheng et al. (2023) proposed including multiple examples

| Source text | Since we do not to bring cash to pay for the transportation fee , enormous time has been saved for everybody . |
|---|---|
| Target text | Since we do not need to bring cash to pay for the transportation fee , enormous time has been saved for everybody . |
| Target edit span | 4 4 need |
| Plain | Since we do not to bring cash to pay for the transportation fee , enormous time has been saved for everybody . |
| System edit span | 4 4 need |

Table 6: Outputs in plain text and edit span formats respectively by LLaMA in the CoNLL2013.

in one prompt and inferring in parallel.

These techniques, unlike our research, do not alter the writing style of the target text, and edit spans can be used in conjunction with these methods.

## 5.2 Edit-based Model

Since the question of necessarily using the seq2seq model for local sequence transduction tasks was raised (Rastogi et al., 2016; Schnober et al., 2016), various edit-based models have been proposed. Guu et al. (2018) proposed a language model that initially selects a prototype sentence from the training dataset and subsequently modifies it to create a new sentence. Ribeiro et al. (2018) introduced a method for representing general string transduction problems as sequence labeling. Koide et al. (2018) proposed the model implemented to analyze the evolution of biological sequences driven by substitution, insertion, and deletion edit operations, achieving improved accuracy on protein secondary structure prediction. Awasthi et al. (2019) presented a parallel iterative edit model reducing decoding time for local sequence transduction tasks. Gu et al. (2019) developed the Levenshtein Transformer, a non-autoregressive model using edit operations. (Mallinson et al., 2020) introduced FELIX, an adaptable text-editing approach for the generation that aims to leverage the advantages of decoding with bi-directional contexts and self-supervised pretraining to the fullest extent. (Xu and Carpuat, 2021) presented an Edit-Based Transformer with Repositioning, which enhances sequence generation flexibility by seamlessly incorporating user-specified preferences in output lexical choice. Reid and Neubig (2022) proposed the modeling of editing processes, encompassing the iterative generation of sequences as a whole. They establish a conceptual framework to explain the probability of multi-step edits and outline neural models capable of learning a generative model of sequences by leveraging these multi-step edits.

However, these methods have different architec-

tures from LLMs. Therefore, it is not easy to apply them to LLMs, unlike our method, which can train models by simply changing the output text.

## 5.3 LLMs for Local Sequence Transduction Tasks

In GEC, the model based on GPT-3 achieves state-of-the-art in unsupervised settings (Loem et al., 2023). Fang et al. (2023) showed that ChatGPT corrects input text very fluently. Yamashita et al. (2020); Rothe et al. (2021); Sun et al. (2022) proposed a method for multilingual GEC using multilingual LLMs. Feng et al. (2023b) investigated the performance of few-shot and zero-shot of GPT3 and ChatGPT in the simplification. Anschütz et al. (2023) used LLMs for German simplification and found them to be effective in languages with little parallel data. (Witteveen and Andrews, 2019) verified the performance of GPT-2 (Radford et al., 2019) in paraphrasing. Wahle et al. (2022) investigated the utilization of T5 and GPT3 in generating machine-generated paraphrases for scientific articles sourced from arXiv, student theses, and Wikipedia. Reif et al. (2022) introduced a method based on GPT-3 that solely relies on natural language instruction and does not necessitate model fine-tuning or exemplars in the desired style. Malmi et al. (2020) proposed a method of using LLMs for style transfer where no parallel data is available. On the other hand, these studies did not target the efficiency of LLMs based on the edit.

## 6 Conclusion

In this study, we proposed to predict a set of edit spans, which represent the changed parts of the target text relative to the source tokens. We showed our method omits unedited tokens that occupy most of the target text and reduces the length of the target text and the inference time for local sequence transduction tasks. Moreover, we reported that instruction tuning with the proposed method achieves

state-of-the-art performance in the four tasks.

## Limitations

In our preliminary experiments, even high-performance LLMs such as GPT-3 (Brown et al., 2020) and ChatGPT (OpenAI, 2023) could not generate edit spans with zero-shot and few-shot. In particular, indexes could not be generated correctly. Therefore, it is a future work to apply the proposed method to zero-shot and few-shot. Moreover, the use of edit span is not necessarily effective for tasks, such as machine translation and dialogue, other than the local sequence transduction task, where many tokens in the source and target texts are not common.

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
