# OpenReview forum: "Reducing Sequence Length by Predicting Edit Spans with Large Language Models"
_EMNLP/2023/Conference — EMNLP 2023 Main_

### Official Review · Reviewer_JGA8 · 2023-08-03

**Soundness:** 4

**Excitement:**

3: Ambivalent: It has merits (e.g., it reports state-of-the-art results, the idea is nice), but there are key weaknesses (e.g., it describes incremental work), and it can significantly benefit from another round of revision. However, I won't object to accepting it if my co-reviewers champion it.

**Missing References:**

* Stahlberg and Kumar,  Seq2Edits: Sequence Transduction Using Span-level Edit Operations, EMNLP 2020
https://aclanthology.org/2020.emnlp-main.418/
* Mallinson et al. EdiT5: Semi-Autoregressive Text-Editing with T5 Warm-Start https://arxiv.org/abs/2205.12209

**Paper Topic And Main Contributions:**

Summary:
This paper proposes the use of edit representations when instruction-tuning/fine-tuning LLMs for localized sequence transduction tasks (e.g. grammatical error correction) where the input and output have substantial overlap. The paper shows that LLMs fine-tuned on edit-span representations for localized transduction tasks and plain text representations for open-ended tasks yield improvements on localized transduction tasks without a large loss in performance on the open ended tasks which use plain text representations. Furthermore, LLMs fine-tuned on task specific data with edit representation, yield large gains in performance relative to existing baselines.

Strengths:
* Proposes a simple approach to fine-tune an LLM on a combination of plain text and edit representations which improves performance on localized sequence transduction tasks without causing a large degradation in open ended tasks with plain text representations
* Demonstrates that using edit representations can substantially reduce the output length of the LLM.
* Shows that LLMs when fine-tuned on task specific data for localized transduction tasks with edit representations can yield good performance relative to existing baselines in the literature.

Weaknesses:
* The idea of using edit representations in sequence-to-sequence models is not novel. See for example: Stahlberg and Kumar,  Seq2Edits: Sequence Transduction Using Span-level Edit Operations, EMNLP 2020
https://aclanthology.org/2020.emnlp-main.418/
* The paper ignores some prior work in editing models e.g. Mallinson et al. EdiT5: Semi-Autoregressive Text-Editing with T5 Warm-Start https://arxiv.org/abs/2205.12209
* There are some confusing statements in the paper - see 'questions to the author' below.


**Questions For The Authors:**

* L186: "To ensure that the token indices do not shift, we apply the edits to the source text in descending order of starting positions". This sentence is confusing - it would be good to clarify this further.
* L254: It would be good to present some examples from the open-ended tasks.
* L300: The phrase: "fine-tuned data" is confusing. Did you mean: "comparing the performance of LLM fine-tuned on plain text vs edit spans"?
* Table 1: The columns should labeled with the metrics.
* L337: "This indicates that the proposed method achieved efficient computational cost during inference without significantly sacrificing open-ended task performance" This sentence is confusing. It would be good to clarify that a) for open-ended tasks, the target is the plain text representation i.e. not the edit spans, b) the savings in computational cost during inference is only applicable to the localized sequence transduction tasks such as grammatical error correction.
* L349: "Table 3 demonstrates the performance difference on … evaluating LLMs trained on both open-ended task and local sequence transduction task data versus LLMs trained solely on local sequence transduction task data". Given the signs of metrics in Table 3, it looks like the above sentence should be "performance difference between LLMs trained solely on local sequence transduction task data (A) and LLMs trained on both open-ended task and local sequence transduction task data (B)". i.e. A-B. The same error is also present in the Table 3 caption.


**Reasons To Accept:**

* Given the popularity of LLMs in the NLP community, the proposed approach is likely to encourage researchers think about alternative output representations which are concise and better suited to the task at hand.
* The paper demonstrates that LLMs can be fine-tuned using a mixture of plain text and edit representations.


**Reasons To Reject:**

* The idea of using edit representations is not entirely novel.
* Some parts of the paper are confusing. See 'questions to the authors'.


**Reproducibility:**

4: Could mostly reproduce the results, but there may be some variation because of sample variance or minor variations in their interpretation of the protocol or method.

**Reviewer Confidence:**

5: Positive that my evaluation is correct. I read the paper very carefully and I am very familiar with related work.

---

> ### Author Rebuttal · Authors · 2023-08-29
>
> > The idea of using edit representations in sequence-to-sequence models is not novel. See for example: Stahlberg and Kumar, Seq2Edits: Sequence Transduction Using Span-level Edit Operations, EMNLP 2020 https://aclanthology.org/2020.emnlp-main.418/
>
> There are two differences between our study and the previous study. First, we demonstrate that the models can generate correct output using only the edit span prompt without changing the architecture. On the other hand, the previous study requires architectural changes, and it is not readily applicable to LLMs. Second, the previous method generates the edit spans for both rewrite and non-rewrite parts. In contrast, our method generates edit spans only for the rewrite part. Consequently, our method has a shorter output sequence length than the previous method. We will reference this study in the camera-ready version and clarify the differences between our study and the previous one.
>
> > The paper ignores some prior work in editing models e.g. Mallinson et al. EdiT5: Semi-Autoregressive Text-Editing with T5 Warm-Start https://arxiv.org/abs/2205.12209
>
> Thank you for sharing the paper. We did not refer to some papers due to the page limit. Therefore, we will add related studies in the final version.
>
> > L300: The phrase: "fine-tuned data" is confusing. Did you mean: "comparing the performance of LLM fine-tuned on plain text vs edit spans"?
>
> Yes, you are right. We rewrite to ``comparing the performance of LLM fine-tuned on plain text vs. edit spans``.

---

### Official Review · Reviewer_R7G8 · 2023-08-05

**Soundness:** 4

**Excitement:**

4: Strong: This paper deepens the understanding of some phenomenon or lowers the barriers to an existing research direction.

**Paper Topic And Main Contributions:**

The authors noted that most source sentences are not transformed in local sequence transformation tasks.
They proposed a method that outputs the modified sentences as tuples instead of outputting them.
The proposed method employs instructional tuning instead of conventional fine-tuning, and the performance was improved in several local sequence transformation tasks.
By fine-tuning each task, SOTA was achieved for GEC, paraphrasing, formal style transformation, and simplification.

**Questions For The Authors:**

A. This tuple consists only of position and inserts words; is there any reason why you did not include labels for insert, replace, and delete?
B. Section 4.2 states that the proposed method can output "none", what is the percentage of "none" in the actual test set?

**Reasons To Accept:**

The authors propose a simpler method than conventional methods that output the entire target sequence.
Such an output-oriented method could be a new baseline for the NLP community, where LLM is becoming mainstream.
In my opinion, such output-oriented methods have various possibilities and should be actively adopted.

**Reasons To Reject:**

- I could not find any merit in shortening the target sequence length.
- (Section 4.1) No performance improvement was observed in 13 of the 32 cases, but the reasons for this were not discussed.

**Reproducibility:**

3: Could reproduce the results with some difficulty. The settings of parameters are underspecified or subjectively determined; the training/evaluation data are not widely available.

**Reviewer Confidence:**

3: Pretty sure, but there's a chance I missed something. Although I have a good feel for this area in general, I did not carefully check the paper's details, e.g., the math, experimental design, or novelty.

---

> ### Author Rebuttal · Authors · 2023-08-29
>
> > I could not find any merit in shortening the target sequence length.
>
> We wrote, ``We can reduce the length of the target text and the inference time for local sequence transduction tasks`` to explain the motivation to reduce the output length in L74-L75 in the introduction section.
>
> > (Section 4.1) No performance improvement was observed in 13 of the 32 cases, but the reasons for this were not discussed.
>
> The performance, especially in style transfer, has decreased. Fig. 2 shows that the style transfer dataset has the longest output sequence when using edit spans. This means that a lot of tokens are rewritten. It is difficult for LLMs that use edit spans to fully benefit from too many edit spans.
>
> > A. This tuple consists only of position and inserts words; is there any reason why you did not include labels for insert, replace, and delete?
>
> The aim of this study is to decrease the output length and make inference time more efficient. The edit positions can represent which edit operation (insert, replace, or delete) has been performed. Therefore, we did not use explicit edit operation tokens to prevent an increase in the output length.
>
> > B. Section 4.2 states that the proposed method can output "none", what is the percentage of "none" in the actual test set?
>
> Only the GEC dataset has ``none``, due to including some texts that do not need to be corrected. For example, 10% of the CoNLL2014 dataset has ``none''.

---

### Official Review · Reviewer_xjk9 · 2023-08-05

**Soundness:** 4

**Excitement:**

3: Ambivalent: It has merits (e.g., it reports state-of-the-art results, the idea is nice), but there are key weaknesses (e.g., it describes incremental work), and it can significantly benefit from another round of revision. However, I won't object to accepting it if my co-reviewers champion it.

**Paper Topic And Main Contributions:**

This paper propose a novel method for applying the large language models to downstream tasks.
The method uses instruction tuning to make LLMs generate edit spans of the source sentence.
Experiments demonstrate that this method performs well on tasks where the source and target sentences are close, outperforms previous methods and achieving significant efficiency improvements.

**Reasons To Accept:**

1. It is a natural idea to use LLMs to generate edit spans for tasks where the source and target sentences are close， which is a simple but effective method.
2. In order to validate this method, the authors conduct solid experiments on multiple tasks.
3. The experimental results demonstrate the superiority of the method.

**Reasons To Reject:**

1. The paper distinguishes text generation tasks into local sequence transduction tasks and open ended tasks, but does not explain the differences between these two series of tasks or cite other work to distinguish them. This may confuse readers who are not familiar with this field.

2. The motivation mentioned in the abstract is that "the models (referring to LLMs according to the context) have a tendency to simply copy the input text as is..." This statement was not mentioned or verified in the subsequent sections. However, to my understanding, LLMs usually perform better than other models on so-called local sequence translation tasks and do not show this tendency.

3. I think reducing the sequence length is a by-product of changing the form of the target, with the biggest benefit being reduced computational cost in generation. So I think the matter of "reducing the sequence length" may not be that important. The comparison of computing costs should be given in the article, such as how much the memory and time are reduced during inference.

**Reproducibility:**

4: Could mostly reproduce the results, but there may be some variation because of sample variance or minor variations in their interpretation of the protocol or method.

**Reviewer Confidence:**

4: Quite sure. I tried to check the important points carefully. It's unlikely, though conceivable, that I missed something that should affect my ratings.

---

> ### Author Rebuttal · Authors · 2023-08-29
>
> > The paper distinguishes text generation tasks into local sequence transduction tasks and open ended tasks, but does not explain the differences between these two series of tasks or cite other work to distinguish them. This may confuse readers who are not familiar with this field.
>
> Local sequence transduction and open-ended tasks: Local sequence transduction tasks have massive overlapping tokens between source and target texts. On the other hand, open-ended tasks don't have such a restriction. We will add this explanation referring to some papers (https://aclanthology.org/D19-1435, https://aclanthology.org/2020.findings-emnlp.136/, https://arxiv.org/abs/2203.02155) in final-version.
>
> > The motivation mentioned in the abstract is that "the models (referring to LLMs according to the context) have a tendency to simply copy the input text as is..." This statement was not mentioned or verified in the subsequent sections. However, to my understanding, LLMs usually perform better than other models on so-called local sequence translation tasks and do not show this tendency.
>
> Thank you for pointing that out. We compare the original LLM and the LLM using the edit spans with the rate of overlapped tokens between source and target texts in order to show LLMs have a tendency to copy tokens in local sequence transduction tasks.
>
> > I think reducing the sequence length is a by-product of changing the form of the target, with the biggest benefit being reduced computational cost in generation. So I think the matter of "reducing the sequence length" may not be that important. The comparison of computing costs should be given in the article, such as how much the memory and time are reduced during inference.
>
> There is a correlation between decreased output length and efficiency. However, we plan to compare the time and memory of the original LLM and LLM using edit spans to show obviously our method is efficient.

---

### Meta-Review · Area_Chair_C4Lr · 2023-09-21

**Recommendation:** 5

**Metareview:**

The paper proposes a method based on instruction tuning of LLMs for localized sequence transduction tasks where source and target sequences have substantial overlap. Experiments demonstrate that model performance was improved in several tasks, such as grammatical error correction, paraphrasing, formal style transformation, and simplification.

Reviewers agree that the method is simple and effective, and that it could possibly be adopted as a new baseline for the NLP community. It was also suggested that such an approach could encourage researchers to think about alternative concise output representations which are better suited to the task at hand.

Some aspects to improve include a comparison of computing costs, such as how much the memory and time are reduced during inference (which the authors' rebuttal agree with and commit to including). In addition, a couple missing references were pointed out.

---

### Decision · Program_Chairs · 2023-10-07

**Decision:**

Accept-Main

**Comment:**

The paper proposes a method based on instruction tuning of LLMs for localized sequence transduction tasks where source and target sequences have substantial overlap. Experiments demonstrate that model performance was improved in several tasks, such as grammatical error correction, paraphrasing, formal style transformation, and simplification.

Reviewers agree that the method is simple and effective, and that it could possibly be adopted as a new baseline for the NLP community. It was also suggested that such an approach could encourage researchers to think about alternative concise output representations which are better suited to the task at hand.

Some aspects to improve include a comparison of computing costs, such as how much the memory and time are reduced during inference (which the authors' rebuttal agree with and commit to including). In addition, a couple missing references were pointed out.